# Drivers and Barriers in Using Industry 4.0: A Perspective of SMEs in Romania

**Mirela Cătălina Türkeş [1],\*, Ionica Oncioiu [2], Hassan Danial Aslam [3], Andreea Marin-Pantelescu [4], Dan Ioan Topor [5] and Sorinel Căpușneanu [2]**

[1]  Faculty of Finance, Banking and Accountancy, Dimitrie Cantemir Christian University, 040051 Bucharest, Romania

[2]  Faculty of Finance-Banking, Accounting and Business Administration, Titu Maiorescu University, 040051 Bucharest, Romania; nelly_oncioiu@yahoo.com (I.O.); sorinelcapusneanu@gmail.com (S.C.)

[3]  Department of Management Sciences, The Islamia University of Bahawalpur, Bahawalpur 63100, Pakistan; hassan.danial@iub.edu.pk

[4]  Faculty of Business and Tourism, The Bucharest University of Economic Studies, 010374 Bucharest, Romania; marin.andreea@com.ase.ro

[5]  Faculty of Economic Sciences, 1 Decembrie 1918 University, 510009 Alba-Iulia, Romania; dan.topor@uab.ro

\*  Correspondence: mirela.turkes@ucdc.ro; Tel.: +40-728-176-475

**Abstract:** Considering the worldwide evolutionary stage of Industry 4.0, this study wants to fill in a lack of information and decision-making, trying to answer a question about the level of preparation of Romanian Small and Medium-sized Enterprises (SMEs) regarding the implementation of the new technology. The main purpose of this article is to identify the opinions and perceptions of SME managers in Romania on the drivers and barriers of implementing Industry 4.0 technology for business development. The research method used in the study was analyzed by sampling using the questionnaire as a data collection tool. It includes closed questions, measured with a nominal and orderly scale. 176 managers provided complete and useful answers to this research. The collected data were analyzed with the Statistical Package for the Social Sciences (SPSS) package using frequency tables, contingency tables, and main component analysis. Major contributions from research have highlighted the fact that Romania is in a full transition process from industry 2.0 to industry 4.0. There was also a high level of knowledge of the new Industry 4.0 technology, and a desire to implement it in the Romanian SMEs, as well as the low level of resources needed to implement it.

**Keywords:** Industry 4.0; SMEs; business; drivers; barriers; managers; digitalization; cloud computing; implementation; flexible manufacturing; systems; cyber-physical systems

## 1. Introduction

IT integration through cyber-physical systems known as Industry 4.0 can be the answer to the issues of addressing the increasing complexity of products and supply chains [1], providing greater flexibility for enterprise-based production systems, and in particular by SMEs. Familiarizing with this concept involves understanding and accepting some holistic and interdisciplinary interpretations or approaches, and many SMEs face certain difficulties. Businesses that integrate advanced technology into production lines will lead to smart grid production lines, and even automation of smart maintenance processes [2]. By reducing costs and failures, the operation of the enterprise will be achieved at the optimum capacity [3].

Some studies by specialists focus on the practices and innovations brought by the Industry 4.0 concept, but also on the expected advantages [4,5], or difficulties encountered [6]. Other specialist studies are focused on analyzing the evolution of the Industry 4.0 concept [7–9] in some industrial branches.

Other studies have highlighted the interests of specialists related to: The digitization of supply chain processes [10], the application of the best strategies for Industry 4.0 [11], selection of the best technologies based on Industry 4.0 [12], the interconnection of logistics services and the promotion of transport cooperation [13], the provision of tools for the selection of maintenance activities with Industry 4.0 [14] and the situation of some countries since the launch of the Industry 4.0 concept [15].

A self-organized, dynamic and real-time infrastructure can be created with the help of these digital transformations, and in this respect, enterprises will be able to achieve their objectives and analyze customer expectations [16]. Knowledge, strategy and planning are the activities which should be incorporated alongside their own technological processes of SMEs. The correct integration of digital processes into a business is based on well-designed planning [17]. Many SMEs face difficulties in achieving these processes and activities. These things are due to the fact that most SMEs do not have the financial resources to invest in new manufacturing technologies and to change the range offered to customers in certain market segments. Monitoring SMEs is difficult when it comes to developing technology and innovation.

SMEs play an important role in the development of the circular economy [18–20] due to the influences of technology and automation in the competitive business environment. SMEs that do not adapt to the requirements of technological evolution become uncompetitive. SMEs are aware of the time in which to learn and adopt new manufacturing technologies that will bring them significant benefits in the future and help them to survive in a globalized environment. In this respect, SMEs in some regions require support, knowledge and training to develop these skills in an efficient manner. Starting from the above-mentioned studies, we intend to conduct a study on the managers' views on the advantages and limitations of the possibilities of implementing Industry 4.0 in SMEs in Romania. According to our opinion, this study will facilitate the transition of SMEs to Industry 4.0 and the results will provide managers with directions for identifying priority criteria, a guide to Industry 4.0 assessment and real possibilities for making appropriate decisions in their business.

This study consists mainly of six parts. In the second part, Industry 4.0 and the components of its paradigms are mentioned in literature, accompanied by its advantages and limitations. In the third part the method used in the quantitative marketing study is presented. In the fourth and fifth parts, there are the results obtained following the application of the method used and their interpretation. In the sixth part, the conclusions and future directions of research are presented.

## 2. Materials

### 2.1. The Fourth Industrial Revolution and Its Paradigms

In its historical evolution, the technology production system used productivity, cost and quality as measuring instruments, all three being closely interconnected. The first Industrial Revolution focused on productivity and manufacturing efficiency as the main axes of the development of manufacturing technology. Improving productivity has been achieved by moving from manual labor to mechanical production. The second Industrial Revolution comes with the introduction of electricity and these results in a significant improvement in productivity. The third Industrial Revolution produces a significant leap by combining IT technology and automation systems (flexible manufacturing systems (FMS) and robotic technology), with work efficiency and productivity. Improving the productivity and flexibility of manufacturing systems continued with the fourth Industrial Revolution (the Internet and smart devices). But what is Industry 4.0? Here is a simple explanation: The Industry 4.0 concept is a branch of material production that integrates innovative elements and technologies (Big Data & Analytics, Simulation of Things-IoT), various devices (Cyber Physical Systems (CPS), Internet of Things, cloud computing) and the functional aspects are approached as services, ensuring a constant communication and relationship [21].

Unlike specialists who define Industry 4.0 as "a new level of value chain organization and product lifecycle management" [22] or "a collective term for technologies and concepts of chain

organization" [23], organizations have defined Industry 4.0 as "integrating complex machines and physical devices with sensors and networking in the network, the software used to predict, control and plan better results in business and in society" [24]. In this way, the Industry 4.0 concept can be perceived as a competitive strategy for the future, but with a strong focus on value chain optimization due to autonomous control and dynamic production. Thus, Industry 4.0 covers the design and implementation of competitive products and services, strong and flexible logistics and production and management systems [25]. The concept of "Industry 4.0" refers to a development that radically changes traditional industries [26], being considered the fourth Industrial Revolution with an extreme impact upon production in the future [27]. According to specialists, production has developed through a new paradigm shift, the so-called "Industry 4.0", where products tend to control their own processing processes [28].

Since the launch of the concept, to date, there are three paradigms that can describe Industry 4.0: Smart product, smart machine and augmented operator. The intelligent product paradigm consists in storing data and individual operational requirements, the system being able to request the necessary resources and coordinate production processes for its completion [29]. The smart machine's paradigm consists in self-organizing machines within a production network, plug-and-play integration, or even replacing new production units, that is, a traditional production hierarchy with a self-organized decentralized CPS [30]. The augmented operator's paradigm transforms the habitual worker into an operator who can intervene manually in an autonomously organized production system based on automated, flexible and adaptive knowledge of the production system [31]. In this way, the operator benefits from mobile support, context-sensitive user interfaces, and user-oriented support systems [32], becoming a strategic decision maker and the ideal solution to solve problems with high technical complexity.

## 2.2. Paradigm Technologies of New Manufacturing

The market and process demands have driven technology from an intense focus of information on an intensive knowledge paradigm, where large data analysis and knowledge bases are key to the current manufacturing environment. The current evolution of integrated and intelligent production processes is driven by both market demand and technological progress. Technologies that can be considered as key elements to support the new manufacturing paradigm are: Integrated manufacturing (3D printing, robotics automation, advanced materials) and intelligent manufacturing (virtual or augmented reality, industrial Internet and CPS), and are based on the following fundamental elements: Large data analysis, cloud computing, applications and mobile devices.

Integrated manufacturing is associated with Cyber Physical Systems (CPS) and IoT (Internet of Things) technologies that have been able to integrate wider manufacturing systems, and are able to process a larger amount of data, information and knowledge at the same time. This process was achieved by horizontal, vertical and end-to-end integration [6,27,33,34]. Vertical integration is linked to intelligent production systems that interconnect all elements of the product lifecycle of an organization (e.g., MES or CAPP used to support the exchange of information and knowledge in SMEs or intelligent factories, smart products, logistics networks, marketing, production or smart services), with a strong focus on needs [6]. Horizontal integration occurs when a company has value-creating networks by integrating suppliers, customers, business partners, business models, or co-operation (for example an industrial Internet that involves a common knowledge platform based on practical protocols and standards necessary to continue to increase efficiency and quality). End-to-End integration involves machine integration and customer integration as parts of the production system, along with product-to-service integration through direct manufacturer monitoring, focusing on product, production and customer design [33].

Three-dimensional (3D) printing is a technology of the future that has been widely embraced by industry, especially aerospace and medical. It plays an important role in product design, research and development and additive manufacturing, contributing to improved production efficiency. Robotic

automation involves the interaction between robots and people that contributes to making easier the material handling processes or some difficult manufacturing situations. To improve production efficiency and reduce costs, automated robots will be used in the future. Due to the advantageous properties they offer, improved materials (carbon fiber, lighter materials with energy storage properties, smart memory), will be widely used in the near future [35].

Intelligent manufacturing is due to increased requirements following the modernization of modern production systems by integrating all the elements needed to make the decisions of a common system. Because there is a need to process large amounts of production data, intelligent capabilities work on sensors, intervene and make decisions with or without human intervention (large data analysis, machine learning, cloud computing). Big Data Analytics involves the process of extracting information and knowledge of large data by discovering hidden clusters and correlations, so that systematic models can be recognized and make better decisions (correlation, statistical modeling and cognitive ML). Machine Learning (ML) refers to the ability of a computer to understand and learn the inside of a physical system through data-based algorithms (data extraction, statistical recognition of algorithm models and artificial neural networks ANN). Cloud computing involves using an Internet-based computing service, that is, sharing a software so that a local user does not have to install that software. Combining emerging technologies such as Cloud Computing, Internet of Things, Service-Oriented Technology, and High Performance Computers makes a new manufacturing platform feasible [36]. The Industrial Internet or IIoT (Industrial Internet of Things) is used for the integration and connectivity of machine and physical devices, people and resources through sensor and software networks for industrial production and operations (a successful IIoT platform is the one called GE: Predix Cloud) [37]. Cyber Physical System (CPS) is a new generation of systems with integrated physical and computing capabilities that can interact with humans in new ways [38], or computers with small sensor networks and actuators that are installed as systems incorporated into materials, equipment and machine components and connected to the Internet [39]. With these devices, local data processing capabilities are created, and useful and abstract information can be communicated through networks, increasing communication efficiency [40,41].

### 2.3. Industry 4.0 Drivers and Limits

The technological revolution was a determining factor for Industry 4.0 [42]. Changes in economic and social life have also led to policies that support industry digitization. People do not have time; everything has to be done quickly and simplified, and modern digital technologies offer this. Thus, High-Tech Strategy is a key element on which Industry 4.0 is built and innovations underlie its development. High-Tech Strategy was launched by the German government in 2006 and was the first national concept to bring together key innovations and stakeholder technology for a common goal of advancing new technologies based on smart services, smart data and cloud computing. Technology offers the advantages of efficiency and competitiveness [43].

Intelligent factories in which people work alongside robots have no longer remained a novelty over the past decades [44]. In Industry 4.0, people are tied to cars and form together a new production system that allows faster and more accurate information exchange. People are another Industry 4.0 driver. An Industry 4.0 driver is highly trained and skilled in the areas of robotics, nanotechnologies, microbiology, or astronautics that communicate with robots and are supported by web technologies and intelligent support systems in their day-to-day activities. Another important driver to support Industry 4.0 is the digitization of production and the elimination of boundaries between the physical world and the digital world [45]. Multiplied and distributed artificial intelligence led to CPS. Production systems have not been designed to replace the abilities and capabilities of human operators, but just to make them more efficient [46].

Industry 4.0's key inductors are represented by the technologies, and digital now acts as a platform to build billions of dollars. Exponential growth of companies such as Uber and Air BnB has taken place thanks to digital. The UK government recently published a report describing its vision for the

revolution of the manufacturing sector by the year 2050. Industry companies such as General Electric (GE), Siemens, ABB and Intel are changing their production strategy and management to embrace Industry 4.0 or the Industrial Internet [47]. The digitization is different from country to country; for example, companies from Japan and Germany are implementing digitization primarily to increase their efficiency and product quality. In the United States, the trend is to develop new business models using digital offers and services, and to provide these products and services digitally as quickly as possible [48]. China's manufacturing companies focus on ways to deal with international competitors by cutting costs [49,50].

Intelligent products (provided with algorithms that can optimize their operations, use and maintenance) and connected (or remotely controlled), are characterized by a high degree of autonomy, being able to function autonomously, self-coordinated and self-diagnosed [51]. Virtual planning systems, simulation models, forecasting, analysis, synthesis, all virtually help production and reduce manufacturing costs [52]. Human-Robot Collaboration, Monitoring and Supervision Support Systems, Digital Computing Assistance Systems and Virtual Training, all are Industry 4.0 drivers [53]. 3G simulation models along with support decision support systems are also factors that encourage and support Industry 4.0 in creating new jobs, contributing to an increase in the competitiveness and productivity of the company's business [54]. There are important changes: In the production chain of new products and services offered to the market [55]; new opportunities (smart products) and new challenges (preparing people to use them); simplification and efficiency of the production process occuring in contrast to the investments for the re-technology of the industry or construction [56]; logistics optimization versus data storage warehouses is achieved. Consumers are the most demanding and focused on Industry 4.0.

Political actors are tools in shaping and supporting Industry 4.0 by including the different players in the speech, and thus creating a common understanding of a future situation in the economic sector, whether health, information and telecommunications, medicine, electrical and electrical engineering, or mechanical engineering [57]. The visualization of Industry 4.0 as an innovative communication capability allows decision-makers to reflect on how they shape innovation in a digital era where communication becomes more and more important. The use of software policy tools such as providing a basis for developing an innovation system with communication actions could become more relevant to the future [58,59]. Industry 4.0 includes three dimensions: High process digitization, intelligent production and connectivity between companies [32]. Analysis and processing of statistical, financial and managerial data is the engine of Industry 4.0. Data analysis tools allow product development and enable companies to expand their services and better align their offers with customer needs [60]. An industry 4.0 development requires super-qualified staff, and this presents challenges for technical colleges and companies that need to join their efforts to prepare new operators in the "factory of the future" [61].

A barrier to the development of Industry 4.0 refers to the ability of machines, computers to oversee the working people, and thus deprives them of intimacy [44], creates dependence on robots, and people become more apathetic, more introverted, saddened, connected to virtual life by seeing an extension of them in the working machine [62]. A new limit in the development of Industry 4.0 is population demography, because we face a negative natural increase in population growth, that is, an aging population. Young people are the people of the future. Where do we get it if we do not have it? We create robots, but who handles them? [63]

Lack of expertise is another barrier to Industry 4.0. (Lack of culture, visions or internal training in the digital domain, as well as lack of specialists are impediments to the accelerated development of Industry 4.0). Impediments to Industry 4.0 development are due to the lack of regulations and working procedures in developing countries, the lack of legislation in place for the development of Cloud Computing, Cyber-Security, Augmented Reality, Artificial Intelligence in Developing Countries [64].

If legislation in Germany, Austria, Switzerland, and the United Kingdom improves and takes into account Industry 4.0 [65] in countries such as Romania, Bulgaria, Hungary, Poland, things are at the beginning.

Industrial Leaders digitize essential activities within their own vertical chain of value and also in their relationship with horizontal partners in the supply chain. In addition, they improve their product portfolio by introducing digital functionality and innovative data services. Worldwide, companies are planning to invest approximately 5% of their digital sales revenues in digitization. These investments will mainly focus on the development of digital technologies such as sensors or connection devices, software and applications such as processing systems. Moreover, companies are investing in employee training and implementing the necessary organizational change. More than half of these companies (55%) consider that they will reimburse these costs within two years. These are the results of PwC's "Industry 4.0: Building the Digital Enterprise" survey, in which over 2000 companies in nine industrial sectors in 26 countries were involved [66].

Digital eco-systems can only work if all participants can trust that their data will not reach the wrong hands. This requires considerable effort from companies, substantial investment in system security and clear data protection standards [67]. Protected Servers, Backup Servers, Secure Servers [68]. Preventing the theft of media, aviation, automotive, technological, and electrical innovation ideas. Stolen prototypes discredit and affect the company's reputation in providing data protection. Hacker cyber-attacks on hospitals are real, and worthwhile cases to consider in the digital age. Thus, in 2017, computers in 74 countries suffered from "ransomware" computer attacks and this led to a state of panic and chaos in the United Kingdom's healthcare system (the National Health System in the United Kingdom), the IT system in Spain and the Telecom system in Portugal.

### 2.4. State of Industry 4.0 in Romania

The state of the implementation of Industry 4.0 in Romania is at the beginning. In the project "Development of the institutional capacity of the Ministry of Economy, SIPOCA code: 7", Industry 4.0 refers precisely to Romania: "Supporting digitization in enterprises in the context of" Industry 4.0, "taking into consideration the overwhelming importance of the EU, in the international trade of Romania". Clusters 4.0 are essential for Romania's competitiveness—the average number of cluster enterprises increased from 10 at the beginning of the clustering process in Romania (2016) to 35 cluster enterprises in 2018. "Clusters in Romania attracted 1962 SMEs with about 110,000 employees and turnover in 2015 reaching 6500 million euros. The export of innovative products for niche markets reached EUR 4670 million in 2015 [69].

In 2018, Romania had funding lines dedicated to the implementation of new innovative concepts and technologies (industry 4.0, 3D Printing and Open innovation) supervised by the Ministry of Economy. The companies in Romania (Deloitte, DHL, DB Schenker, Volvo, Microsoft, Oracle, Honeywell) invest in Transportation Management System, Warehouse Management System, Mobile Devices, BI software) and applications that help track orders and supply chain transparency. But the biggest companies are at the forefront of implementing Internet of Things (IoT) or augmented reality (AR) solutions. For example, the development of 4G and 5G networks will lead to a revolution in IoT. In 2018, mobile phones were outnumbered by IoT devices.

Taking into account the evolutionary stage of Industry 4.0 in Romania, we intend to carry out this study to highlight managers' views on the advantages and limitations offered by Industry 4.0 in SMEs. We believe that this study will be particularly useful to managers and specialists in all fields interested in implementing Industry 4.0 in Romania, as well as to other countries that are in transition to Industry 4.0.

## 3. Research Methodology

Given the increased importance of implementing Industry 4.0 technologies that drive business growth, a study was conducted amongst SMEs in Romania. The decision-making issue delineated in

the study follows "How prepared are Romanian SMEs to implement Industry 4.0 technology?" Eligible for SME status are all enterprises that meet the following three criteria: The number of employees is less than 250 persons; the enterprise has an annual turnover not exceeding EUR 50 million; or it has an annual balance sheet total not exceeding EUR 43 million [70].

By focusing on the typology of the technologies employed, on the drivers, and on the benefits generated by and tracking the views and perceptions of SME managers on the implementation of these digital business development technologies, we set the following objectives:

O1.    *Knowing the Industry 4.0 concept and identifying the need for partnerships to implement specific technologies;*
O2.    *Identification of types of technologies to be implemented by enterprises and their level of training;*
O3.    *Assessing enterprise drivers following the use of Industry 4.0 Technology;*
O4.    *Determining the barriers that businesses might encounter in implementing Industry4.0 technologies;*
O5.    *Link between employees and developing Industry 4.0 technologies.*

In order to meet the above objectives, data was collected from enterprises that meet the following size criteria: Micro-enterprises with 0–9 employees, small enterprises with 10–49 employees and medium enterprises with 50–249 employees. All these companies operate in Romania in areas such as auto, pharmaceutical, industrial, IT, chemical, consulting, electronics, insurance, oil and gas and health. The study was conducted between 15 October and 15 December 2018.

The population surveyed and relevant for the study was identified in the metadata database of the National Institute of Statistics in Romania (INSSE). First, the database provided us with the information needed to define and build the surveyed community (sampling base) based on criteria such as enterprise size and industries where technology is applicable. The process ended with the creation of a cross-list of 2750 businesses. In a second step, due to some errors (such as: Inactive, non-contact, dissolving or already dissolved), the sampling base (list) was reduced to 1500 businesses. After identifying the enterprises, many contacts with their representatives were established both for the acquisition of the sample survey agreement and for the knowledge of the managers who were to receive the questionnaire by e-mail and to complete it later. The research method used in the quantitative study was the survey by sampling, using the questionnaire as a data collection tool. It included closed questions, measured with nominal and ordinal scales.

The series of questions included in the questionnaire are as follows: (1) The knowledge of the concept "Industry 4.0" by the management of the enterprise; (2) looking or not looking for solution partners to help implement Industry 4.0 technologies; (3) identification of the types of technology to be implemented by enterprises; (4) establishing the general training level of the enterprise for their implementation; (5) assessing enterprise drivers using industry 4.0; (6) determining the barriers that companies might encounter in implementing Industry 4.0 technology; (7) building an expert team within the enterprise; (8) training employees to use digital tools for remote collaboration and connectivity; and (9) encouraging employees to propose new ideas for technology deployment within the enterprise.

The study was carried out with the support of 12 interviewers who all had experience in this area. To increase the quality of research results, and to reduce the impact of errors due to e-mail communication, the 12 operators have been pre-trained, receiving all the necessary instructions for carrying out the research in optimum conditions. Operators sent a link to the survey questionnaire by email to managers who agreed to participate in the survey. Between 15 October and 15 December 2018, several reminders were sent to them. The process ended with the conclusion of participation agreements with the 604 companies included in the list. The answer rate was 29.1%, i.e., 176 managers provided complete and useful answers for this research. The collected data was analyzed with the SPSS package using different bivariate methods, such as: Frequency tables, contingency tables and main component analysis (regression method).

## 4. Results and Discussions

The first research objective (O1) consisted in knowing the concept of Industry 4.0 by enterprise management and looking, or not looking, for solution partners to help implement the industry 4.0 technologies. As can be seen in Table 1, 8 out of 10 SME managers have knowledge of the fourth Industrial Revolution, while 15.9% of them say that this concept has been launched in Romania only at the declarative level. 88.9% of business managers with between 50 and 249 employees are studying how to use the business growth concept appropriately, and the managements of 81.2% of small businesses claim that in recent years "Industry 4.0" has emerged as a necessity to streamline production processes.

**Table 1.** Contingency table for knowing the Industry 4.0 concept in relation to enterprise size.

| Cross Tabulation | | | Number of Employees | | | Total |
| --- | --- | --- | --- | --- | --- | --- |
| | | | Between 0 and 9 Employees | Between 10 and 49 Employees | Between 50 and 249 Employees | |
| Have you heard of the Industry 4.0 concept? | No | Count | 15 | 9 | 4 | 28 |
| | | % without number of employees | 16.3% | 18.8% | 11.1% | 15.9% |
| | Yes | Number | 77 | 39 | 32 | 148 |
| | | % without number of employees | 83.7% | 81.2% | 88.9% | 84.1% |
| Total | | Number | 92 | 48 | 36 | 176 |
| | | % without number of employees | 100.0% | 100.0% | 100.0% | 100.0% |

Research results indicate that 23.9% of SME managers are looking for partners to: Implement information and communications technology; to digitize information and integrate systems into product conception, development, manufacturing and use; to adopt new software technologies for modeling, simulation, virtualization and digital manufacturing; and for the development of cyber-physical systems to monitor and control physical processes. 26.1% of SMEs already have partnerships with specialists to use Cloud, Big Data or to design some types of autonomous robots that will contribute to product development, from prototype and zero series to serial production. Managers of 22.7% of SMEs state that their employees have skills for using and communicating with the 3G/4G mobile networking technology; for operating with Android, BlackBerry OS, webOS, iOS, Symbian, Windows Mobile Professional, Windows Mobile Standard and Bada smartphones; to use smart devices that can detect temperature or light, and more. Nearly 27.3% of SME's are still hesitating in search of technology partners in Industry 4.0 (Table 2 or Figure 1).

**Table 2.** Contingency table on searching for solution sartners for implementing technologies according to industry types.

| Cross Tabulation | | | Industry | | | | | | Total |
|---|---|---|---|---|---|---|---|---|---|
| | | | Automotive | Pharma | Oil/Gas | Chemical | Electronics | Other Industries | |
| Looking for solution partners to help you implement Industry 4.0? | Yes | Number | 9 | 4 | 1 | 0 | 2 | 26 | 42 |
| | | % out of total | 5.1% | 2.3% | 0.6% | 0.0% | 1.1% | 14.8% | 23.9% |
| | No, we found a partner | Number | 11 | 4 | 3 | 1 | 1 | 26 | 46 |
| | | % out of total | 6.2% | 2.3% | 1.7% | 0.6% | 0.6% | 14.8% | 26.1% |
| | No, we already have the competence | Number | 10 | 6 | 3 | 3 | 5 | 13 | 40 |
| | | % out of total | 5.7% | 3.4% | 1.7% | 1.7% | 2.8% | 7.4% | 22.7% |
| | Hesitating | Number | 12 | 5 | 0 | 8 | 2 | 21 | 48 |
| | | % out of total | 6.8% | 2.8% | 0.0% | 4.5% | 1.1% | 11.9% | 27.3% |
| Total | | Number | 42 | 19 | 7 | 12 | 10 | 86 | 176 |
| | | % out of total | 23.9% | 10.8% | 4.0% | 6.8% | 5.7% | 48.9% | 100.0% |

or

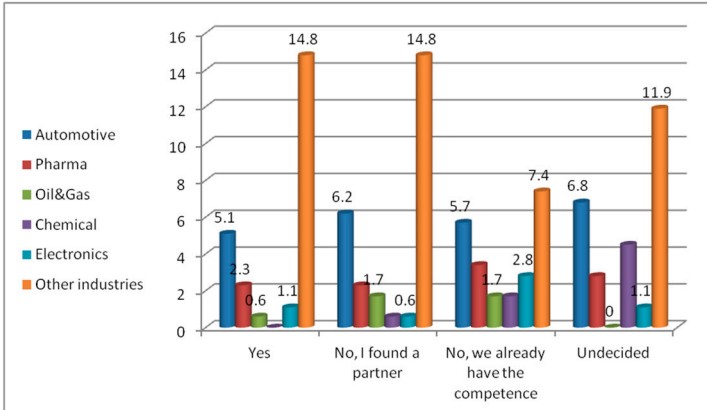

**Figure 1.** Distribution of answers regarding the search for solution partners for implementing technologies in relation to industry types.

While nine SMEs in the automotive industry are looking for partners to integrate new technologies, 12 entities are undecided, and 21 units either have found partners or employees with skills in the field. SMEs operating in industries such as pharmaceuticals, electronics, chemicals and oil and gas seem less interested in concluding contracts with solution partners, and more geared towards searching for and building the best digital talents.

The second research objective (O2) involved identifying the types of technologies to be implemented in the future by enterprises and their level of training.

Table 3 shows that the managers of the 176 SMEs indicated 352 responses. The modal value was recorded for the "Autonomous Robots" technology, with 62 responses, representing 17.6% of the total responses. This technology is to be implemented in the future by 35.2% of SMEs responding to this question. Other technologies indicated by SMEs as necessary to be implemented for product

and product development were: "Horizontal & Vertical System Integration" (27.8%), "Big Data & Analytics" (27.3%), "Simulation" of Things (IoT) "(21.6%) and" Cyber-Security "(17.6%).

The least responses were obtained by Artificial Intelligence (1.7%) technology, to be implemented in the future by only (3.4%) SMEs.

**Table 3.** Frequency table for multiple response analysis regarding future technology types implemented by Small and Medium-sized Enterprises (SMEs).

| $V3 Frequencies | | Answers | | Percentage of Cases |
|---|---|---|---|---|
| | | N | Percent | |
| Types [a] | Big Data & Analytics | 48 | 13.6% | 27.3% |
| | Autonomous Robots | 62 | 17.6% | 35.2% |
| | Simulation | 38 | 10.8% | 21.6% |
| | Horizontal & Vertical System Integration | 49 | 13.9% | 27.8% |
| | Internet of Things (IoT) | 38 | 10.8% | 21.6% |
| | Cyber-Security | 31 | 8.8% | 17.6% |
| | Additive Manufacturing | 20 | 5.7% | 11.4% |
| | Augmented Reality | 14 | 4.0% | 8.0% |
| | Cloud Computing | 13 | 3.7% | 7.4% |
| | Mobile Technologies | 24 | 6.8% | 13.6% |
| | Artificial Intelligence | 6 | 1.7% | 3.4% |
| | Radio-Frequency Identification (RFID) & Real-time locating system (RTLS) technologies | 9 | 2.6% | 5.1% |
| Total | | 352 | 100.0% | 200.0% |

Figure 2 shows the training level of SMEs for the implementation of Industry 4.0 specific technologies. Level 0 was indicated by 72.2% of SMEs, which have not been involved in the Industrial Revolution so far. Level 1 was reached by 32 SMEs (18.2%), while Level 2 was achieved by 8.5% (15) of enterprises. Level 3 was only indicated by two enterprises having between 50 and 249 employees, with managers indicating that they are in the process of implementing digitization to increase the efficiency and quality of their own products. None of the 176 SMEs reached levels 4 and 5 yet. However, 20.5% of SME managers having between 50 and 249 employees state that in future, digitization from one end to the other of all physical assets and processes, as well as integration into digital ecosystems with partners in the value chain, must be realized (Table 4).

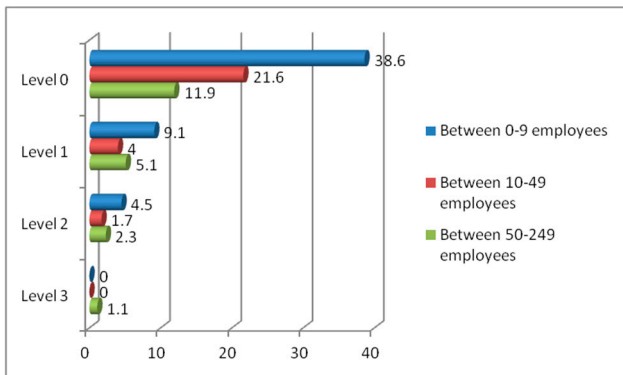

**Figure 2.** Distribution of answers regarding the level of training of SMEs in the implementation of Industry 4.0 specific technologies.

or

**Table 4.** The contingency table regarding the training level of SMEs in relation to the size of enterprises.

| Crosstabulation | | | Number of Employees | | | Total |
|---|---|---|---|---|---|---|
| | | | Between 0 and 9 Employees | Between 10 and 49 Employees | Between 50 and 249 Employees | |
| What is the enterprise's general training level for implementing Industry 4.0 technology? | Level 0 | Count | 68 | 38 | 21 | 127 |
| | | % of Total | 38.6% | 21.6% | 11.9% | 72.2% |
| | Level 1 | Count | 16 | 7 | 9 | 32 |
| | | % of Total | 9.1% | 4.0% | 5.1% | 18.2% |
| | Level 2 | Count | 8 | 3 | 4 | 15 |
| | | % of Total | 4.5% | 1.7% | 2.3% | 8.5% |
| | Level 3 | Count | 0 | 0 | 2 | 2 |
| | | % of Total | 0.0% | 0.0% | 1.1% | 1.1% |
| Total | | Count | 92 | 48 | 36 | 176 |
| | | % of Total | 52.3% | 27.3% | 20.5% | 100.0% |

The third research objective (O3) implied the assessment of enterprise drivers following the use of Industry 4.0 technology. Using the analysis of the main components, the interdependencies between five variables based on which SMEs in Romania expressed their agreement or disagreement on the drivers resulting from the implementation of Industry 4.0 technologies, were studied.

Table 5 shows that for the 176 SMEs included in the analysis there were relatively different but close approximations for the five variables. For the "Customer requirements" and "Competitors Practice Industry 4.0" variables, both the median and the modal value are 2, which is the "Disagree" response. "Neutral" was the answer given by SME managers to the "To reduce costs", "To improve time to market" and "Due to legal requirements" variables, and revealed by the median and modal values, this response took the value 3.

**Table 5.** Descriptive statistics.

| | Mean | Median | Mode | Std. Deviation | Analysis N |
|---|---|---|---|---|---|
| Customer requirements | 2.16 | 2.00 | 2 | 1.053 | 176 |
| Competitors practice Industry 4.0 | 2.22 | 2.00 | 2 | 1.094 | 176 |
| To reduce costs | 3.19 | 3.00 | 3 | 3.21 | 176 |
| To improve time to market | 2.98 | 3.00 | 3 | 1.119 | 176 |
| Due to legal requirements/changed legislation | 2.74 | 3.00 | 3 | 1.183 | 176 |

Table 6 shows that there is a relatively high correlation between the "Customer Requirements" and "Legal Requirements/Amended Legislation" appraisals as well as the "Industry Practitioners 4.0" and "Legal Requirements/Amended Legislation" appraisals. A small correlation is found between the "Reduce Costs" and "Competitors Practice Industry 4.0".

**Table 6.** Correlation coefficient matrix of the 5 analyzed variables.

| Correlation Matrix | | Customer Requirements | Competitors Practice Industry 4.0 | Reduce Costs | Market -Time-Improvement | Legal Requirements/ Amended Legislation |
|---|---|---|---|---|---|---|
| Correlation | Customer Requirements | 1.000 | −0.013 | −0.020 | −0.129 | 0.110 |
| | Competitors Practice Industry 4.0 | −0.013 | 1.000 | 0.023 | −0.021 | 0.112 |
| | Reduce Costs | −0.020 | 0.023 | 1.000 | −0.017 | 0.026 |
| | Market-time improvement | −0.129 | −0.021 | −0.017 | 1.000 | −0.051 |
| | Legal Requirements/ Amended Legislation | 0.110 | 0.112 | 0.026 | −0.051 | 1.000 |

Figure 3 shows the graphical representation of the association between the variables and the main components. The first component is determined by "Legal Requirements/Amended Legislation" (0.623), "Customer Requirements" (0.616) and "Competitors practicing Industry 4.0" (0.351). The second component was determined by the following variables: "Competitors practicing Industry 4.0" (0.678), "Reduce costs" (0.428) and "Market-time improvement" (0.348).

Figure 4 shows that the attitudes of medium enterprises are towards the second component, and the attitudes of micro and small enterprises towards the first component. The attitudes of medium-sized enterprises were positive towards "Legal Requirements/Amended Legislation" and negative to "Customer requirements". For microenterprises and small businesses, there is a positive attitude towards the "Legal Requirements/Amended Legislation" and negative towards "Cost Reduction" and "Competitors Who Practice Industry 4.0".

The fourth research objective (O4) was to determine the barriers that companies could encounter after implementing Industry 4.0 technology. The study of the interdependencies between the 6 variables was based on the analysis of the main components, the managers of the analyzed SMEs expressing their agreement or disagreement about the barriers that could be generated after the implementation of Industry 4.0 technologies.

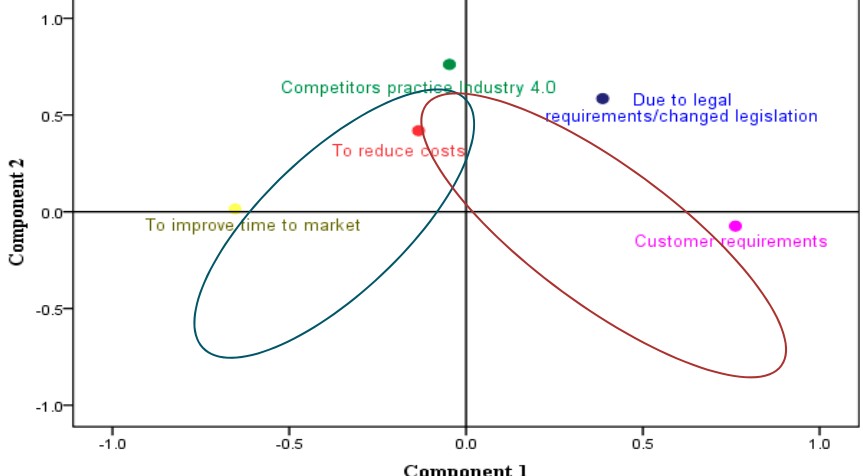

**Figure 3.** Graphical representation of the association between the 5 variables and the main components.

The averages obtained by the 176 SMEs analyzed were different in size for all six variables. "Lack of standards" was the only variable that obtained the value 2 for both the median and the modal value.

The average score recorded by the variables: "Lack of knowledge about Industry 4.0" (3.55), "More focus on operation at the expense of developing the company" (3.34), Lack of understanding of the strategic importance of Industry 4.0 "(3.27) "Too few human resources (3.19)," Requires continued education of employees "(3.03), ranged between" Neutral "and" Agree "levels (Table 7).

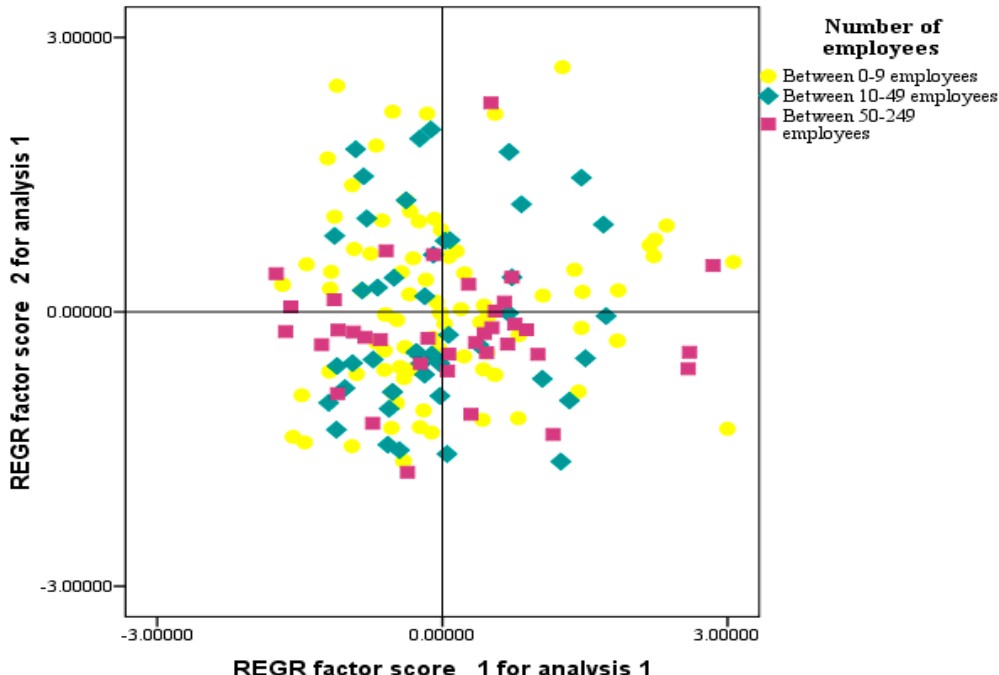

**Figure 4.** Representation of SMEs by size in the two main components.

**Table 7.** Matrix of correlation coefficients between the 6 variables analyzed.

|  | Mean | Median | Mode | Std. Deviation | Analysis N |
|---|---|---|---|---|---|
| Lack of knowledge about Industry 4.0 | 3.55 | 3.00 | 3 | 1.304 | 176 |
| Lack of standards | 2.94 | 2.00 | 2 | 1.177 | 176 |
| More focus on operation at the expense of developing the company | 3.34 | 3.00 | 3 | 1.093 | 176 |
| Lack of understanding of the strategic importance of Industry 4.0 | 3.27 | 3.00 | 3 | 1.225 | 176 |
| Too few human resources (man power) | 3.19 | 3.00 | 3 | 1.158 | 176 |
| Requires continued education of employees | 3.03 | 3.00 | 3 | 1.11 | 176 |

As can be seen in Table 8, there are three small correlations between "Lack of knowledge about Industry 4.0" and "Too few human resources" (0.117), "Lack of knowledge about Industry 4.0" and "Lack of understanding of the strategic importance of Industry 4.0 "(0.094), and between" More focus on operating costs and the need for continuing education of employees "(0.085). In addition to the above, the lowest correlation encountered was the appreciation of "More focus on the operation of the company" and "Lack of understanding of the strategic importance of Industry 4.0" (0.001).

**Table 8.** Matrix of correlation coefficients between the 6 variables analyzed.

| Correlation Matrix | | Lack of Knowledge about Industry 4.0 | Lack of Standards | More Focus on Operation at the Expense of Developing the Company | Lack of Understanding of the Strategic Importance of Industry 4.0 | Too few Human Resources (Man Power) | Requires Continued Education of Employees |
|---|---|---|---|---|---|---|---|
| Correlation | Lack of knowledge about Industry 4.0 | 1.000 | −0.078 | −0.057 | 0.094 | 0.117 | 0.023 |
| | Lack of standards | −0.078 | 1.000 | −0.072 | −0.044 | 0.063 | 0.050 |
| | More focus on operation at the expense of developing the company | −0.057 | −0.072 | 1.000 | 0.001 | −0.068 | 0.085 |
| | Lack of understanding of the strategic importance of Industry 4.0 | 0.094 | −0.044 | 0.001 | 1.000 | −0.104 | 0.002 |
| | Too few human resources (man power) | 0.117 | 0.063 | −0.068 | −0.104 | 1.000 | 0.039 |
| | Requires continued education of employees | 0.023 | 0.050 | 0.085 | 0.002 | 0.039 | 1.000 |

The graphical representation of the association between the six variables and the main components is shown in Figure 5. The first component is determined by "Too few human resources" (0.718), "Lack of standards" (0.469) and "Lack of knowledge about Industry 4.0") and "Requires continued education of employees" (0.082). The second component was determined by the following variables: "Lack of knowledge about Industry 4.0" (0.784), "Lack of understanding of the strategic importance of Industry 4.0" (0.503) and "Too few human resources".

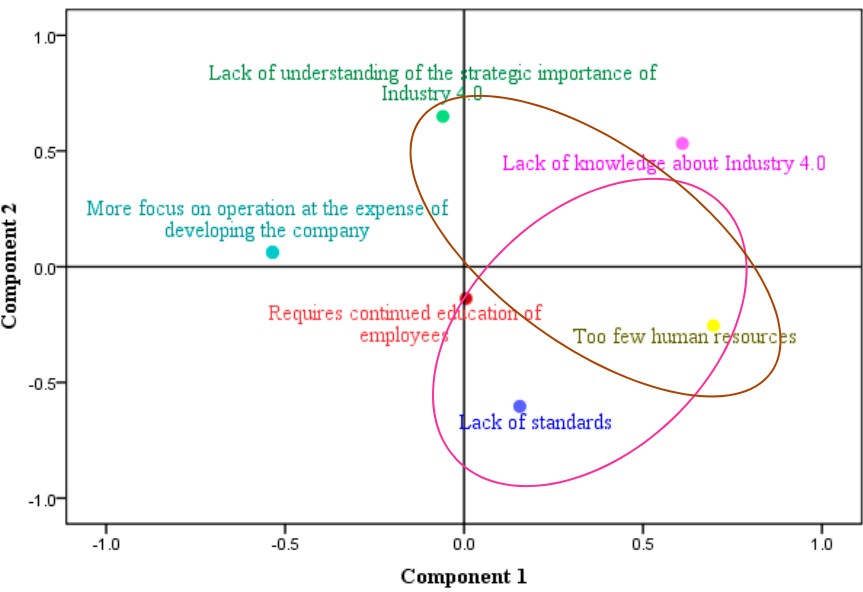

**Figure 5.** Graphical representation of the association between six variables and the main components.

Figure 6 shows that the positive attitudes of "Pharma" and "Oil and Gas" companies to "Lack of knowledge about Industry 4.0" and the "Automotive" and "Chemical" entities to "Lack of the understanding of the strategic importance of Industry 4.0 ". For enterprises in "Other Industry" there is a negative attitude towards "Requires continued education of employees" and "Lack of standards".

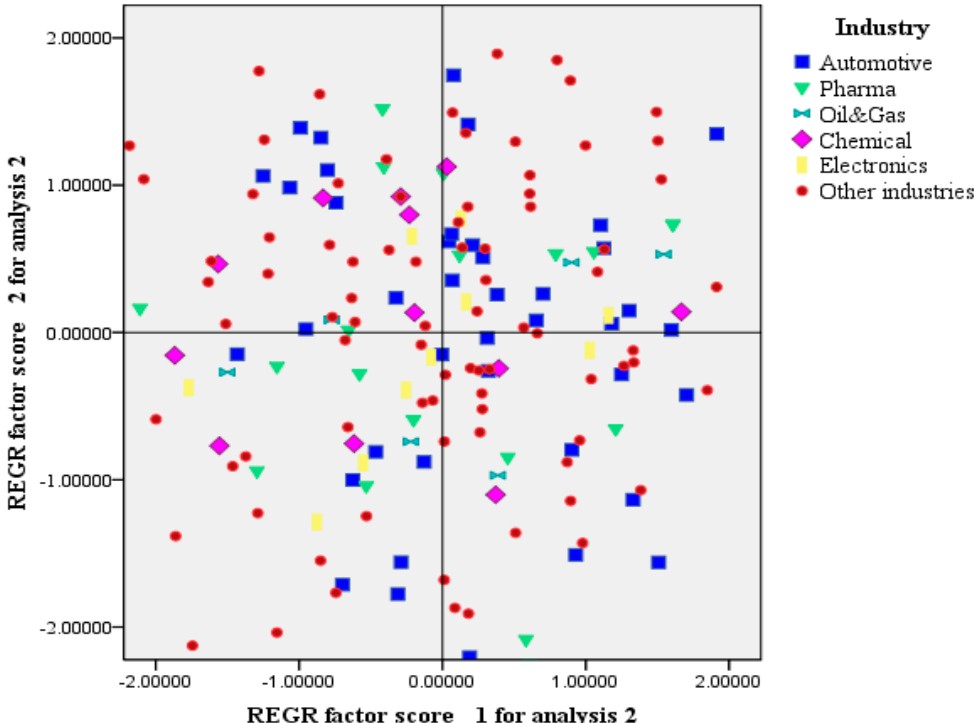

**Figure 6.** SME representation by industry in terms of the two main components.

The fifth research objective (O5) was to establish a link between SME employees and the development of Industry 4.0 technologies. Table 9 shows the extent to which SMEs have built digitization expert teams with company-level responsibilities for strategy and industry-specific implementation of Industry 4.0 technology. Out of the 176 respondents SMEs, only 30.7% of entities have organized expert teams for business digitization, while the remaining 122 enterprises have not done so. Two-thirds (2/3) of businesses with between 50 and 249 employees have already formed expert teams on digitization, while 8 out of 9 micro-enterprises (between 10 and 49 employees) have not built such a team.

**Table 9.** Table of contingency on the extent to which SMEs have built digitization expert teams in relation to the size of the enterprise.

| Crosstabulation | | | Number of Employees | | | Total |
|---|---|---|---|---|---|---|
| | | | Between 0 and 9 Employees | Between 10 and 49 Employees | Between 50 and 249 Employees | |
| Have you built a team of experts in the enterprise to implement Industry 4.0 technologies? | No | Count % of Total | 82 46.6% | 29 16.5% | 11 6.2% | 122 69.3% |
| | Yes | Count % of Total | 10 5.7% | 19 10.8% | 25 14.2% | 54 30.7% |
| Total | | Count % of Total | 92 52.3% | 48 27.3% | 36 20.5% | 176 100.0% |

Table 10 shows the extent to which SMEs train their own employees to become experts in using digital tools for remote collaboration and connectivity. Only 56.2% of SMEs have trained their employees, while 43.8% of entities have not yet developed an annual budget for training or other refresher courses in digitization. Only 69.4% of medium-sized enterprises (between 50 and

249 employees) have already allocated resources to increase digital skills, the remaining 11 entities have not undertaken such actions. The situation is different among micro-enterprises (between 0 and 9 employees), where 66.3% of entities do not see an obstacle in the absence of the annual budget for specialized staff or the impossibility of training them in the digital domain.

**Table 10.** The contingency table on the extent to which SMEs train their own employees in digitization in relation to the size of the enterprise.

| Crosstabulation | | | Number of Employees | | | Total |
|---|---|---|---|---|---|---|
| | | | Between 0 and 9 Employees | Between 10 and 49 Employees | Between 50 and 249 Employees | |
| Have you trained your employees to use digital tools for remote collaboration and connectivity? | No | Count % of Total | 61 34.7% | 27 15.3% | 11 6.2% | 99 56.2% |
| | Yes | Count % of Total | 31 17.6% | 21 11.9% | 25 14.2% | 77 43.8% |
| Total | | Count % of Total | 92 52.3% | 48 27.3% | 36 20.5% | 176 100.0% |

Table 11 shows the extent to which enterprise management encourages employees' ideas to implement Industry 4.0 technologies. 9 out of 10 managers encourage their employees to spend between 20% and 30% of their working time on innovative and creative projects related to the development of digital technologies at the enterprise level. Employees from micro-enterprises (47.2%) and small enterprises (23.9%) are encouraged, while medium-sized staff are less supported by management (19.3%).

**Table 11.** The contingency table regarding the extent to which enterprise management encourages employees' ideas to implement Industry 4.0 technologies in relation to enterprise size.

| Crosstabulation | | | Number of Employees | | | Total |
|---|---|---|---|---|---|---|
| | | | Between 0 and 9 Employees | Between 10 and 49 Employees | Between 50 and 249 Employees | |
| Does enterprise management encourage employees' ideas for implementing Industry 4.0 Technologies? | No | Count % of Total | 9 5.1% | 6 3.4% | 2 1.1% | 17 9.7% |
| | Yes | Count % of Total | 83 47.2% | 42 23.9% | 34 19.3% | 159 90.3% |
| Total | | Count | 92 | 48 | 36 | 176 |

## 5. Conclusions

This study assessed the views and perceptions of business managers on the future implementation of new technologies to better understand the challenges and opportunities of digitization in the SME sector, and how it can contribute to business development.

More than 84% of SME managers have knowledge of the fourth Industrial Revolution, while nearly 16% of them say "No," but I think the shift from 2.0 to 4.0 Industry requires a lot of determination, a high capital for the adoption of new technologies, and a well-trained human capital in digitization. Research results indicate that 23.9% of SME managers are looking for partners to implement new technologies, and 26.1% of SMEs have already signed up with specialists in this field. Another 22.7% of managers have the skills to use the new technologies needed in their own field of activity, while the remaining 27.3% have declared themselves undecided.

According to the managers' opinions, the most important technologies to be implemented in the future by the Romanian SMEs are: "Autonomous Robots" (35.2%), "Horizontal & Vertical System Integration" (27.8%), "Big Data & Analytics" (21.6%), "Internet of Things (IoT)" (21.6%) and "Cyber-Security" (17.6%).

As far as the level of preparation for the implementation of Industry 4.0 specific technologies is concerned, 72.2% of the analyzed SMEs indicated level 0, 18. 2% level 1, 8.5% level 2, while level 3 was

indicated by only 2 enterprises which have between 50 and 249 employees. As levels 4 and 5 were not reached by any of the 176 investigated SMEs, we understand that the entities being analyzed are not yet ready to face the leap from Industry 2.0 to Industry 4.0. The managers of 20.5% of SMEs with between 50 and 249 employees believe that such actions should be initiated, and through efficient collaboration and putting together all the resources (skills, technology and digitization) they can create a healthy business ecosystem, and sustainable economic growth can be ensured.

Research results indicate that SMEs, regardless of their size, are considering the following drivers: Lack of knowledge about Industry 4.0 (3.55); lack of standards (2.94); more focus on operation at the expense of developing the company (3.34); lack of understanding of the strategic importance of Industry 4.0 (3.27); too few human resources (man power) (3.19); requirement of the continued education of employees (3.03).

Given the barriers to industry 4.0, SMEs, regardless of their industry, agree that they will face "Lack of knowledge about Industry 4.0" (3.55), "More focus on operation at the costs of developing the company" (3.34), "Lack of understanding of the strategic importance of Industry 4.0" (3.27), "Too few human resources" (3.19), "Requires continued education of employees" (3.03), and "Lack of standards"(2.94).

The fact that businesses are not ready to leap to Industry 4.0 demonstrates the extent to which SMEs have built expert teams with responsibilities in the strategy and activities specific to the implementation of new technologies. Out of the 176 respondent SMEs, only 30.7% of entities have organized expert teams for business digitization, while the remaining 69.3% of entities have done nothing in this respect. The same situation is confirmed by the extent to which SMEs train their employees to become experts in the use of digital tools. Only 56.2% of SMEs have organized digitization trainings and refresher courses. However, 90.3% of SME managers encourage their employees to allocate between 20%–30% of their working time to projects aimed at implementing digital technologies. If medium-sized enterprise management encourages employees less than 19.3%, micro and small enterprises show a higher rate of encouragement, at 47.2% and 23.9%, respectively.

We must recall the limitations of this research. The first limitation of this study is related to the size of the sample, reduced in size, and this due to a low response rate, which leads to a self-selection process of the respondents (SMEs). However, we must not forget that the answers obtained are of significant relevance to the objectives of the study, and provide new research directions that lead to overcoming these limitations. Secondly, it should be recalled that the data on the population surveyed were obtained from the National Institute of Statistics (INSSE) metadata database. Other studies may lead to different outcomes and conclusions in terms of the objectives analyzed in this study. The opinions and perceptions of managers in other countries or other areas may be extremely different.

**Author Contributions:** I.O. and H.D.A. Conceptualization and supervision; M.C.T. and S.C. writing-original draft preparation, methodology, writing—review and editing; A.M.-P. and D.I.T. writing—review and editing.

**Funding:** This research received no external funding.

**Conflicts of Interest:** The authors declare no conflict of interest.

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
