# Peer review of "Drivers and Barriers in Using Industry 4.0: A Perspective of SMEs in Romania"

_processes, doi:10.3390/pr7030153_

Round 1

Reviewer 1 Report

              Review of the Manuscripts ID processes-460577 „Drivers and Barriers in Using Industry 4.0: A Perspective of SMEs in Romania” for the Journal Processes.

              The paper „Drivers and Barriers in Using Industry 4.0: A Perspective of SMEs in Romania” intends to   fill in a lack of   information      I and decision-making, trying to answer a question about the level of preparation of Romanian SMEs regarding the implementation of the new technology.

              The paper consists of five sections: Introduction, Materials, Research methodology, Results and Discussion and Conclusions.

              The abstract contains the main purpose of the paper, the research method used in the research and the main contributions.

              It would be very useful to add in the "Introduction" section the purpose, objectives and hypothesis of the research. At the end of this part, the authors mistakenly mention that the research contains six parts. In fact, the paper contains five parts.

              The paper contains the second section titled Materials, which in fact represents literature review. We recommend authors to quote recently published papers on SMEs in the Sustainability and Current Science journal, namely: Batrancea, I .; Morar, I.-D .; Masca, E .; Catalin, S .; Bechis, L. Econometric Modeling of SME Performance. Case of Romania. Sustainability 2018, Volume 10, Issue 1, Article Number: 192, DOI: 10.3390 / su10010192, Jan.2018, ISSN: 2071-1050; Gaban L., Masca E., Morar I.D, Fatacean Gh., Moscviciov A., Statistical Analysis of Performance in SMEs, Current Science, Vol. 115, Issue 8, pp.1543-1550, ISSN: 0011-3891.

              In the third part “Research methodology”, the authors present how SMEs have been selected based on Romanian regulations. Also, in this part are mentioned the objectives of the research.

In the fourth part „Results and disussions”, the authors explain results of the databased of research using the SPSS package using frequency tables, contingency tables, and main component analysis.

              In “Conclusion” part of the paper, the authors explain their findings regarding the advantages and the limits using industry 4.0 in Romanian SMEs. The results of the research presented in the paper are valuable and should be presented to the scientific audience in the economic theory. 

Author Response

Hello,

We have analyzed your reviews. thanks to them, we have had our chance to improve and complete our work.

Thanks once again for your esteem reviews.

Yours Sincerely,

Mirela Turkes

Reviewer 2 Report

Dear Authors

This paper is very interesting work, emphasizing how important for the Industry 4.0 concept is to meet the individual needs of customers. However, I have some Major comments.

Strengths:

1.   An interesting partition of SMEs in Romania.

2.   Experimental results are provided for evaluation.

Weaknesses:

1.  Reason behind deploying in a Drivers and Barriers should be elaborated more. Given a irregular Industry 4.0, how would the proposed system deployed with Industry 4.0?

2.  The paper should be conducted proof-reading.

3.  Important references are missing:

1) Drivers and barriers affecting usage of e-Customs—A global survey with customs administrations using multivariate analysis techniques." Government Information Quarterly 30.4 (2013): 473-485.

2) A Study on Improvement of Blockchain Application to Overcome Vulnerability of IoT Multiplatform Security. Energies, 12(3), 402.

3) An Optimized Algorithm and Test Bed for Improvement of Efficiency of ESS and Energy Use. Electronics, 7(12), 388.

4) Drivers and barriers of extended supply chain practices for energy saving and emission reduction among Chinese manufacturers." Journal of Cleaner Production 40 (2013): 6-12.

5) Barriers and drivers for sustainable building. Building Research & Information, 39(3), 239-255.

6) "Drivers and barriers of sustainable design and construction: The perception of green building experience." International Journal of Sustainable Building Technology and Urban Development 4.1 (2013): 35-45.

7) A Cost-Effective Redundant Digital Excitation Control System and Test Bed Experiment for Safe Power Supply for Process Industry 4.0. Processes, 6(7), 85.

8) Effect of Cooperation on Manufacturing IT Project Development and Test Bed for Successful Industry 4.0 Project: Safety Management for Security. Processes, 6(7), 88.

4. What is Industry 4.0 ?
I would like to explain it in the paper.

5. What is SMEs ?
I would like to explain it in the paper.

Author Response

Hello,

We have analyzed your reviews. thanks of them, we have had our change to improve and complete our work.

Thanks once again for your esteem reviews.

We request a proofreading of our article.

Yours Sincerely,

Mirela Turkes

Round 2

Reviewer 2 Report

Dear Authors

The revision adequately address the concerns expressed in last review.

I recommend that this revised paper "Accept in present form".